# Aggregation-Caused Quenching Dyes as Potent Tools to Track the Integrity of Antitumor Nanocarriers: A Mini-Review

**DOI:** 10.3390/ph18020176

**Published:** 2025-01-27

**Authors:** Xiye Wang, Jiayue Huang, Mengqin Guo, Yiling Zhong, Zhengwei Huang

**Affiliations:** State Key Laboratory of Bioactive Molecules and Druggability Assessment, Guangdong Basic Research Center of Excellence for Natural Bioactive Molecules and Discovery of Innovative Drugs, College of Pharmacy, Jinan University, Guangzhou 510006, China; wxy8848wxy@stu2022.jnu.edu.cn (X.W.); jnu2022102959@stu2022.jnu.edu.cn (J.H.); guomengqin@stu2021.jnu.edu.cn (M.G.)

**Keywords:** nanocarrier, structural integrity, aggregation-caused quenching, dyes, antitumor, drug delivery

## Abstract

Cancer has become one of the major causes of death worldwide. Chemotherapy remains a cornerstone of cancer treatment. To enhance the tumor-targeting efficiency of chemotherapy agents, pharmaceutical scientists have developed nanocarriers. However, the in vivo structural integrity and dynamic changes in nanocarriers after administration are not well understood, which may significantly impact their tumor-targeting abilities. In this paper, we propose the use of environmentally responsive fluorescent probes to track the integrity of antitumor nanocarriers. We compare three main types of dyes: fluorescence resonance energy transfer (FRET) dyes, aggregation-induced emission (AIE) dyes, and aggregation-caused quenching (ACQ) dyes. Among them, ACQ dyes, possessing sensitive water-quenching properties and easily detected “on–off” switching behavior, are regarded as the most promising choice. We believe that ACQ dyes are suitable for investigating the in vivo fate of antitumor nanocarriers and can aid in designing improved nanoformulations for chemotherapy agents.

## 1. Introduction

According to the latest cancer statistics published in CA: A Cancer Journal for Clinicians, lung cancer was the most frequently diagnosed cancer in 2022, accounting for one in eight cancer cases worldwide (12.4% of all cancers globally). Other leading cancers by incidence included female breast cancer (11.6%), colorectal cancer (9.6%), prostate cancer (7.3%), stomach cancer (4.9%), liver cancer (4.3%), thyroid cancer (4.1%), cervical cancer (3.3%), bladder cancer (3.1%), and non-Hodgkin lymphoma (2.8%). At the same time, lung cancer remained the leading cause of cancer death, responsible for approximately 1.8 million deaths. Other cancers with high mortality rates included colorectal cancer, liver cancer, stomach cancer, breast cancer, pancreatic cancer, esophageal cancer, prostate cancer, cervical cancer, and leukemia [1]. These cancers not only lead to significant loss of life but also consume substantial economic, medical, and human resources [2]. In addition to the frequently reported cancer types mentioned above, there are many other cancer types that are receiving increasing clinical attention. Bone cancer, for example, is nowadays a tumor with an increasing incidence among young people.

Currently, surgical resection remains the primary treatment for many solid tumors. However, issues such as postoperative complications, decreased quality of life, tumor recurrence, and metastasis can affect the outcomes of surgical treatment [3]. Radiotherapy is utilized in more than 60% of cancer cases and inhibits cancer cell growth by causing DNA damage. However, it can also harm nearby normal cells, leading to adverse effects such as secondary malignancies and infertility [4]. In addition to surgery and radiotherapy, chemotherapy is a common treatment modality for cancers (Figure 1). The German chemist Paul Ehrlich coined the term “chemotherapy”, defining it as the use of chemicals to treat diseases [5]. Chemotherapy for cancer began in the 1940s, and, since then, many chemotherapeutic agents have been developed, including doxorubicin [6], pemetrexed [7], gefitinib [7], and cisplatin [8]. However, conventional formulations of chemotherapy agents often face challenges with poor tumor targeting due to the lack of a specific targeting mechanism [9]. As a result, treatments may be less effective and are often accompanied by systemic toxicity and side effects [10]. Therefore, efforts should be made to enhance the tumor targetability of chemotherapy agents.

The field of nanomedicine has witnessed unprecedented growth in the past few years, with continuous experimentation and testing leading to the development of modern nanostructures for the diagnosis and treatment of diseases, in particular, cancer [11]. Currently approved cancer nanomedicines are predominantly liposomal formulations and drug conjugates (proteins, polymers, and/or antibodies), focusing on improving the pharmacokinetics and pharmacodynamics (PK/PD) of free drugs and utilizing passive targeting [12]. The mechanisms of passive targeting are preferentially due to enhanced permeability and retention effects (EPR) [13]. Targeting drug delivery is one of the main areas in which nanotechnology continues to fundamentally change the course of cancer treatment. Currently, nanostructures are being developed in two main ways with different modes: First, the nanostructures themselves can be used as both carriers and chemotherapeutic agents [14]. Second, drugs can be absorbed directly into or dissolved within the nanoparticle framework or covalently bound to the surface of the nanostructures [15]. According to the above two modes, the nanostructures can be used to deliver drugs to the body as a carrier or as a chemotherapeutic agent *per se* [11].

With advances in nanotechnology, pharmaceutical scientists have developed nanocarriers to load chemotherapy drugs, thereby establishing nanoformulations that can achieve tumor-targeted drug delivery [16]. Various types of nanocarriers have been reported, including albumin-based carriers [17], liposomes [18], magnetic carriers [19], and polymeric carriers [20]. Utilizing this approach, multiple chemotherapy nanoformulations—such as Abraxane, Genexol, and Paclical—have successfully reached clinical application [21]. These nanocarriers can achieve tumor targeting through both passive and active mechanisms. Passive targeting relies on physicochemical or pharmacological factors of the disease to guide therapeutic agents into target tissues [22]. In contrast, active targeting involves modifying carriers with ligands that bind specifically to receptors on tumor cells, thereby significantly enhancing drug delivery efficiency compared to passive targeting [23]. The use of nanoparticles in chemotherapy is not limited to drug delivery. In many cases, nanoparticles were used to reduce the toxicity of chemotherapeutic drugs or to control drug release. Gold nanoparticles (AuNPs), for example, are ideally suited as diagnostic agents and for therapeutic use due to their unique properties, such as modulatable surface plasmon resonance (SPR), high surface reactivity, biocompatibility, and energy-absorbing capacity [24,25]. Studies have shown that AuNPs are toxic and quite difficult to manufacture. It was demonstrated that the use of botanical phytochemicals in nanobiotechnology to encapsulate AuNPs can lower the toxicity and increase feasibility for large-scale production [26]. Similarly, DNA nanoparticles (NPs) can also be used as carriers to wrap drugs for controlled release purposes. The carrier is suitable for wrapping gene and protein drugs and can protect the drugs from enzyme degradation. Specific targeting can also be achieved in vivo by changing the DNA structure, and the DNA nanoparticles (NPs) can also use their characteristics specific to changes in temperature, light, and pH to achieve controlled release. DNA nanoparticles (NPs) show potential for treating multiple diseases [27]. Overall, chemotherapy drug formulations based on nanocarriers hold considerable promise for improving cancer treatment outcomes.

For bone cancer, mentioned above, researchers assessed the antitumor potential of a newly developed phosphonic acid derivative (2-carboxyethylphenylphosphonic acid) using two different in vitro human cell cultures (keratinocyte-forming cells HaCaT and osteosarcoma SAOS-2 cells), employing different techniques (MTT reagent [3-(4,5-dimethylthiazol-2-yl)-2,5-diphenyltetrazolium bromide], cytomorphometric evaluation, lactate dehydrogenase (LDH), Hoechst staining, and real-time reverse transcription polymerase chain reaction (RT-PCR)), outlining its pharmacological profiles. It was found that the compounds had good biocompatibility and excellent biosafety, and their anti-osteosarcoma activity was high. Therefore, the newly developed compounds should be considered as promising candidates for further in vitro and in vivo studies related to the treatment of osteosarcoma. Some phosphorus-containing organic–inorganic hybrids were demonstrated to have considerable positive effects on malignant cells of human osteosarcoma [28]. This phosphorus-containing organic–inorganic hybrid can be synthesized by the sol-gel method, and Merning’s team found that hybrid materials were synthesized by the modification of Ti(OPr)4 by diphenyl phosphinic (DPPA) or phenyl phosphonic acid (PPA), making the material very stable [29]. It was shown that phosphorus-containing organic–inorganic hybrids may have a positive effect on malignant cells of human osteosarcoma. The study used a variety of techniques to determine the anticancer characteristics of 2-carboxyethylphenylphosphinic acid and found that the drug has good anti-osteosarcoma activity and safety in normal cells compared to the three phosphorus-containing compounds on the market. Typically, hybrid compounds containing phosphorus can be prepared by a sol-gel process that utilizes the ability of phosphate groups to pair with metal oxides in organic components. The composition and structural stability of the products prepared by this process were also verified [30].

This mini-review aimed to discuss how the limited understanding of the in vivo fate of nanocarriers affects the therapeutic efficacy of chemotherapeutic nanoformulations, including factors such as nanocarrier stability and tumor-targeting capacity. In comparison with other existing monitoring techniques, we propose using aggregation-caused quenching (ACQ) dyes—characterized by their sensitive water-quenching properties and easily detectable “on–off” switching behavior—to track the integrity of antitumor nanocarriers in vivo following administration. A conceptual framework for this review is illustrated in Figure 2. We envision that harnessing the distinguishability of ACQ dyes to explore the in vivo fate of antitumor nanocarriers may help overcome current developmental bottlenecks in nanocarrier-based chemotherapy.

## 2. Major Strategies for Developing Antitumor Nanocarriers

Nowadays, drug incorporation methods for nanoparticles have become abundant and mature. Traditional incorporation methods include using liposomal nanoparticles to encapsulate drugs [30], which selectively bind to target cells and release the drugs; using magnetic nanoparticle-targeted delivery systems combined with the action of an external magnetic field to precisely deliver drugs to specific organs and tissues [31]; encapsulating nucleic acid drugs in lipid nanoparticles; followed by surface modification with specific ligands or antibodies, the lipid nanoparticles are able to target cells or tissues and release the drugs [32]. We can integrate these methods into two dominant domains, viz., to control drug release and improve cellular uptake. Hereby, this mini-review will briefly summarize the representative cases of controlling drug release and increasing permeability into cancer cells.

### 2.1. Controlling Drug Release

Controlling the drug release of antitumor nanocarriers can be achieved by applying external stimuli and taking advantage of the nanocarrier nature, which is introduced below.

#### 2.1.1. Applying External Stimuli

Distinguished from unresponsive antitumor nanocarrier designs, we believe that the design of applying external stimuli to control drug release is novel and effective. In contrast to endogenous stimuli in biological environments, the responsiveness of nanoparticles to exogenous stimuli allows for remote control of the action at specific points in space and time. Once acting externally on the target tissue, the exogenous stimulus can be easily controlled after loading a drug to a specific nanoparticle. This ensures that rationally designed nanoparticles can be activated at the demanded time and site. It was evidenced that the design approach of using external stimuli to facilitate drug release is feasible [33]. For example, 2 nm nanoparticulate gold clusters with good heating effect under radiofrequency electric field (RF-EF) irradiation were used for the in vitro thermotherapy of cancer cells. The researchers prepared doxorubicin liposomes and exposed the prepared formulations to RF-EF. The experimental results demonstrated that 20.2 ± 2.1% of the drug was released and the IC50 value of colorectal cancer cells decreased twofold. The X-ray attenuation efficiency of liposomal gold clusters was superior to that of commercial iodohexitol and free gold clusters at different concentrations. Finally, treatment of cancer cells with gold clusters significantly reduced cell survival under cobalt-60 beam irradiation [34].

Typical features of hypoxic cells, such as low oxygen levels and highly bioreductive environments, may provide stimulus-responsive drug release to aid in tumor-specific chemotherapy, radiotherapy, photodynamic therapy, and acoustic kinetic therapy. Experimental results demonstrated the successful development of hypoxia-responsive nanocarriers (tumor microenvironment (TME) receptive and modulating therapeutic nano drug delivery systems (DDS)) that could be used for drug delivery to heterogeneous tumors. This approach of targeting hypoxic tumor habitats was expected to overcome difficulties arising from tumor heterogeneity and could be used to design diagnostic and therapeutic nanocarriers to target various types of solid tumors [35].

#### 2.1.2. Taking Advantage of Nanocarrier Nature

The triggering of drug release can be designed directly by utilizing the inherent properties of the nanocarrier material; however, it should be noted that a stimulus is still needed in this scenario. Confronting the challenge that nanomaterials suffer from insufficient delivery of therapeutic drugs at the target site, in recent years, researchers have developed stimuli-responsive nanocarriers [36,37,38,39,40], which change their properties when subjected to various stimuli, in order to deliver drugs and genes to the target site in a controlled and adequate manner. The classical examples include stimuli-responsive polymers for antimicrobial therapy [41], a DNA delivery nanocarrier (called Pluronic-PEI-SS, synthesized by conjugating reducible disulfide-linked PEI (PEI-SS) with Pluronic, was fabricated [42]), enzyme-responsive liposomes [43], and stimuli-responsive nanocarriers [44].

Of note, stimuli-responsive nanocarriers usually contain various environmentally sensitive features in their structures, so that the loaded therapeutic drugs can be released in response to various environmental factors, such as endogenous stimuli (e.g., temperature, pH, redox potential, enzymes, etc.) and exogenous stimuli (e.g., electromagnetism, light, radiation, ultrasound, etc.) [45]. By this means, versatile nanocarriers can be developed.

### 2.2. Increasing the Permeability into Cancer Cells

Similar to controlling drug release, increasing the permeability into cancer cells of antitumor nanocarriers can be achieved by applying external stimuli and taking advantage of nanocarrier nature as well.

#### 2.2.1. Applying External Stimuli

Increased accumulation of nanoparticles in tumors using the EPR effect is known as a passive targeting approach [46], which is widely recognized. However, relying solely on the EPR effect may be insufficient for tumoral accumulation, and external stimuli should be introduced. In a study using folate-functionalized gold magnetic core-shell nanostructures for the treatment of HPV (human papillomavirus)-positive oral cancer, HPV-positive oral cancer cells were subjected to nanoparticles, electric fields, and radiation treatments. Tumor cell viability and apoptosis rates were determined, and cellular uptake of the nanoparticles was also determined by inductively coupled plasma-optical emission spectrometer (ICP-OES) analysis. It was found that none of the treatments alone resulted in significant cancer cell death. Combined treatments increased the mortality of cancer cells and increased the proportion of apoptotic cells among them. It was also observed that the electric field enhanced the uptake of the nanoparticles by the cancer cells, and it could be concluded that the combination of folic acid-functionalized nanoparticles and electroporation opened up a new avenue for improving the efficacy of radiation therapy for human papillomavirus-positive cancers. This finding is integral to the application of external stimuli to increase the permeability of the nanoparticles into the cancer cells [47].

#### 2.2.2. Taking Advantage of Nanocarrier Nature

Novel strategies such as transformable nanocarriers, transcellular delivery of peptide-modified nanocarriers, and bio-inspired carriers have recently emerged as a new generation of drug carriers with uptake enhancement possibilities [46]. For example, polymeric nanoparticles (like Adriamycin-encapsulated nanoliposomes with polyethylene glycol (PEG) coating enhance the circulating half-life of the drug required for the treatment of cancer (Kaposi’s sarcoma, recurrent ovarian cancer, etc.) [48]. These nanoparticles have a small size range and variable shape. Their small size allows them to penetrate and be taken up by cells, thereby increasing the accumulation of the drug at the targeted tumor site. Several methods can be used to incorporate drugs into polymeric nanoparticles, such as dissolution, precipitation, adsorption, or attachment) [49,50,51,52]. Albumin nanoparticle carrier systems were designed to take advantage of the presence of different drug binding sites in the albumin molecule and the fact that a large number of drugs can be doped into the particle matrix [53]. Albumin nanoparticles are easy to prepare, smaller in size, biodegradable, non-toxic, harmless, easy to metabolize, and are very suitable natural polymers [54].

## 3. The Developmental Bottleneck for Antitumor Nanocarriers

It is encouraging that chemotherapy agent formulations based on nanocarriers have made progress, as reflected by several commercially available products, including ThermoDox^®^ and Genexol^®^PM [55]. However, beyond these examples, practical successes remain limited, and many nanoformulations have not achieved successful clinical translation [56]. Thus, the real-world application of nanocarrier-based chemotherapy still requires significant improvement.

A primary bottleneck is the limited understanding of the in vivo fate of nanocarriers within the academic community, particularly regarding the dynamic changes in their structural integrity after administration. Developing nanocarriers guided by predictions of their in vivo behavior has become a prevailing trend, especially in the context of formulation research and development driven by “Quality by Design” (QbD) [57]. Nonetheless, despite this emerging focus, our knowledge of the in vivo structural integrity of nanocarriers remains scarce. This integrity is linked not only to their physical and chemical stability but also to their tumor-targeting capacity. Notably, if the targeting ligand on a nanocarrier is compromised before reaching the tumor site, the carrier loses its targeting function and cannot achieve tumor-targeted drug delivery [58]. Consequently, the therapeutic advantages of nanocarriers cannot be realized, diminishing their role in tumor therapy. In such a scenario, formulations intended to be “targeted” become indistinguishable from non-targeted ones and may accumulate in non-tumor tissues, potentially causing unfavorable outcomes [59]. This can result in insufficient therapeutic efficacy and an increased risk of systemic toxicity, further burdening patients. Moreover, nanocarriers’ interactions with biomolecules in vivo, along with their degradation and release characteristics, may also be affected [8]. These factors collectively influence the targeting, effectiveness, and safety of anticancer therapeutics. Without sufficient knowledge of the in vivo structural integrity of nanocarriers, the scientific community cannot effectively guide the research, development, and application of antitumor nanocarriers. This gap in understanding thus stands as a significant obstacle to progress in this field.

To achieve this goal, it is crucial to closely monitor the structural integrity of nanocarriers during the development of chemotherapeutic nanoformulations. Such monitoring not only affects the stability and drug release efficiency of nanocarriers in vivo but also significantly influences the eventual success of targeted therapy. To overcome current bottlenecks, we must adopt a systematic approach to monitoring nanocarrier integrity. First, comprehensive fundamental research is needed to understand how different nanocarriers maintain stability and degrade under various physiological conditions. This involves examining their responses to biological environments, interactions with biomolecules, and metabolic processes in vivo. Second, from a methodological perspective, it is essential to develop and implement a set of advanced monitoring techniques capable of assessing nanocarrier integrity in real time. Although the theoretical aspects were thoroughly discussed in several previous reviews [60], the methodological considerations remain less explored. By providing insights into these cutting-edge monitoring technologies, we aim to facilitate their translation into practical clinical applications, thereby advancing the development of antitumor nanoformulations and ultimately improving the quality of life for cancer patients.

Current monitoring technologies, such as fluorescence labeling, isotope labeling, and direct content determination, face a critical limitation: they lack the ability to distinguish between intact and structurally compromised nanocarriers. For instance, in commonly used fluorescence and isotope labeling methods, both intact (labeled) and damaged (unlabeled) nanocarriers can produce similar signals once the labels are released, making it impossible to differentiate between them. Similarly, direct content determination methods typically rely on extracting all nanomaterials—whether intact or not—using organic solvents and measuring them collectively. As a result, the observed signal is likely a mixture of intact and partially degraded nanocarriers in unknown proportions [61]. Consequently, the findings obtained from these techniques may not accurately reflect the true structural integrity of the nanocarriers, potentially leading to artifacts and misinterpretation.

In order to clarify the dynamic changes in the structural integrity of nanocarriers in vivo, a strategy capable of distinguishing signals from intact and non-intact nanocarriers should be employed, thereby providing reliable data for a more in-depth analysis of their behavior in vivo.

## 4. Aggregation-Caused Quenching (ACQ) Dyes: A Promising Solution

Early in the development of nanomedicine, a plausible strategy to achieve “discrimination” was proposed: non-degradable labeling methods [62]. In these approaches, a marker (e.g., fluorescent or isotopic) is either physically incorporated into a non-biodegradable, rigid nanocarrier or chemically grafted to it via a strong covalent bond. Under such labeling schemes, all detected signals originate from intact nanocarriers, seemingly solving the distinction problem. However, non-degradable labeling methods are not suitable for clinical use. Non-biodegradable materials tend to accumulate in the body, and, given the limited metabolic enzyme activity in vivo, this accumulation can raise serious safety concerns [63,64]. Consequently, there is an urgent need to develop new strategies that are both safer and capable of providing the required “distinguishability”.

Bioimaging based on environmentally responsive fluorescent probes is a promising new strategy to achieve “distinguishability”. Unlike conventional fluorescent probes, environment-sensitive probes respond to changes in the local microenvironment [65]. In the context of in vivo fate studies, these changes primarily refer to the structural integrity of nanocarriers—specifically, whether the probes remain encapsulated within an intact nanocarrier or were released into an aqueous biological matrix due to a compromised structure. As previously mentioned, conventional fluorescent probes emit strong signals regardless of whether they are contained in an intact nanocarrier, as shown in Figure 3 (left panel). In contrast, environmentally responsive fluorescent probes display distinct fluorescence characteristics (e.g., shifts in wavelength or intensity) depending on whether they are located in intact or incomplete nanocarriers. This allows researchers to differentiate between the two states, as illustrated in Figure 3 (right panel) [61]. In this case, imaging electrode heterogeneity using chemically constrained fluorescence electrochemical microscopy enables visualization of the interaction regions of fluorescent probes [33].

To date, three main categories of environmentally responsive probes have been introduced: aggregation-induced emission (AIE) probes [66], fluorescence resonance energy transfer (FRET) probes [67], and aggregation-caused quenching (ACQ) probes [68], as illustrated in Figure 4.

FRET probes consist of pairs of fluorescent molecules with overlapping emission spectra, enabling dipole–dipole interactions based on the FRET phenomenon [70]. FRET involves the transfer of radiant energy from a donor molecule in the excited state to an acceptor molecule in the ground state when excited by an appropriate wavelength of light [71]. This process occurs only if the donor and acceptor molecules are within approximately 10 nm of each other. In an intact nanocarrier, the spatial constraints maintain the donor–acceptor pair within this critical distance, allowing FRET to occur. Conversely, when the nanocarrier is disrupted and the probes disperse into an aqueous biomatrix, the distance between the molecules exceeds 10 nm, causing the FRET effect to be lost. Thus, FRET probes can produce distinct fluorescence signals under intact versus disrupted conditions, providing a means to distinguish nanocarrier integrity. Common FRET probe pairs include FITC (Fluorescein Isothiocyanate)/TRITC (Tetramethylrhodamine) (Figure 5A) [72], Cy3/Cy5(Figure 5B), and FAM (FAM alkyne,6-isomer)/BHQ-2 (BHQ-2 amine) (Figure 5C) [73]. Their excitation wavelengths are 495 nm/547 nm, 550 nm/650 nm, and 495 nm/nm emission, respectively, while their emission wavelengths are 519 nm/572 nm, 570 nm/670 nm, and 520 nm/nm emission, respectively.

AIE and ACQ probes represent two sides of the same coin, as both types of probes feature large, conjugated aromatic structures that aggregate in aqueous biological environments due to hydrophobic interactions (e.g., π–π stacking) [74].

Herein, the detailed aggregation mechanisms are as follows. AIE and ACQ probes are both organic molecules with large conjugate structures, hence owning a high hydrophobicity. Water, in most cases, is a non-solvent for these molecules. As a result, the intermolecular interactions will take place among probes, rather than between probe and water, in an aqueous environment. In accordance with the theory of molecular exciton coupling, the plane-to-plane hypsochromically shifted hypsochromic aggregate (H-aggregate) and the end-to-end bathochromically shifted jelly-aggregate (J-aggregate) will generate when the probe monomers make contact with each other in a non-solvent [69]; both H-aggregate and J-aggregate manifest in the pattern of aggregation [11].

The key difference lies in their molecular flexibility. AIE probes, with rigid molecular structures, form aggregates that open radiative channels, allowing energy to be released as fluorescence. In contrast, ACQ probes, possessing more flexible structures, form aggregates that open nonradiative channels, leading to energy dissipation without fluorescence emission [75]. Consequently, AIE probes exhibit strong fluorescence upon aggregation, while ACQ probes undergo fluorescence quenching under the same conditions. When these probes are released from disintegrated nanocarriers into aqueous environments, AIE probes emit bright fluorescence, whereas ACQ probes become quenched. Thus, both types of probes can serve as “discriminatory” indicators of the nanocarrier’s structural integrity. Common AIE probes include Tetraphenylethylene (TPE) (Figure 6D), poly(aroxycarbonyltriazole)s (PACT) (Figure 6E), and TPE-BT-BBTD (Figure 6F), with excitation/emission wavelengths of 350 nm/450 nm, 390 nm/510 nm, and 380 nm/490 nm, respectively [76,77]. On the ACQ side, probes P1 through P4 (Figure 7) differ in their conformational restrictions within aza-BODIPY or BODIPY structures, resulting in excitation/emission wavelengths of 740 nm/752 nm, 708 nm/732 nm, 688 nm/715 nm, and 651 nm/662 nm, respectively [68,76,78,79].

All three types of environmentally responsive probes possess varying degrees of “discriminatory power”, but the extent of this capability differs. In the case of FRET probes, they consistently emit a strong fluorescence signal, regardless of whether the FRET effect occurs. While the fluorescence wavelength may shift depending on the distance between probe molecules, this shift is often minimal and may not be readily distinguishable [77]. For AIE probes, fluorescence is enhanced in aqueous biological matrices; however, this enhancement can be influenced by factors such as viscosity, concentration, and temperature [82]. In some instances, these factor-dependent enhancements are limited [83], thus imposing more stringent conditions on their application. As a result, current fluorescence analyzers—especially small animal in vivo imaging systems without adjustable detection windows—may struggle to differentiate between intact and non-intact nanocarriers when using FRET or AIE probes.

In contrast, ACQ probes fluoresce (“on”) when confined within intact nanocarriers and quench (“off”) upon release into an aqueous environment if the nanocarriers are compromised. This “on/off” transition is so pronounced that standard fluorescence analyzers can easily detect it. Consequently, among these three categories of probes, ACQ probes theoretically possess the strongest “discriminatory power” and are well-suited for monitoring the structural integrity of antitumor nanocarriers.

The molecular structures of the aforementioned dyes are depicted in Figure 5, Figure 6 and Figure 7, which can be consulted for further application.

## 5. Concluding Remarks

Ultimately, we concluded that ACQ dyes, in theory, provide the highest sensitivity and “resolution” compared to FRET and AIE dyes, whose fluorescent signals cannot effectively distinguish between intact and incomplete nanocarriers. Utilizing ACQ dyes can therefore help elucidate the dynamic changes in nanocarrier structural integrity in vivo, thereby facilitating the research and development of nanocarrier-based chemotherapy formulations. In recent years, researchers have employed ACQ dyes with an aza-BODIPY skeleton to investigate the in vivo fate of various nanocarriers. For example, one study used these dyes to determine nanocarrier integrity in vivo and to explore the relationship between particle size and lung retention time after pulmonary administration [84]. Another study tracked the fate of fat emulsion droplets to differentiate various nutritional fat emulsion products [85]. Additionally, ACQ dyes were applied to assess the in vivo behavior of polymer nanoparticles administered orally [86]. They also showed excellent performance in probing the in vivo fate of antitumor nanocarriers, such as nitric oxide (NO)-propelled nanomicelle-integrated dissolving microneedles [59]. Together, these findings demonstrate that ACQ dyes possess remarkable discriminatory power and can be widely adopted in the field.

Although the future development of fluorescent probes is very promising, in the real-world process of application, the properties of fluorescent probes may be affected by the physiological environment and other various reasons. For quick examples, the reactive oxygen monitoring probe will be affected by various chemical interactions and may not be specific [87] or, in the process of nitrite detection in the body, the fluorescent probe may become not sensitive enough [88]. These shortcomings still encourage us to carry out continuous studies and find suitable strategies to circumvent them.

Looking ahead, we plan to conduct a series of in-depth studies to comprehensively showcase the unique advantages of ACQ dyes in tracking the structural integrity of antitumor nanocarriers. We intend to systematically design experiments to evaluate their labeling and monitoring capabilities under diverse conditions. Through these investigations, we aim to highlight the potential of ACQ dyes in improving the accuracy of visualizing and monitoring antitumor nanocarriers. In addition, we will establish a technological platform to monitor nanocarriers delivered via less commonly studied administration routes—such as transdermal, pulmonary, and nasal pathways—to gain a clearer understanding of their behavior, distribution, and release mechanisms in vivo. This knowledge will ultimately provide more precise guidance for developing future tumor treatment strategies.

## Figures and Tables

**Figure 1 pharmaceuticals-18-00176-f001:**
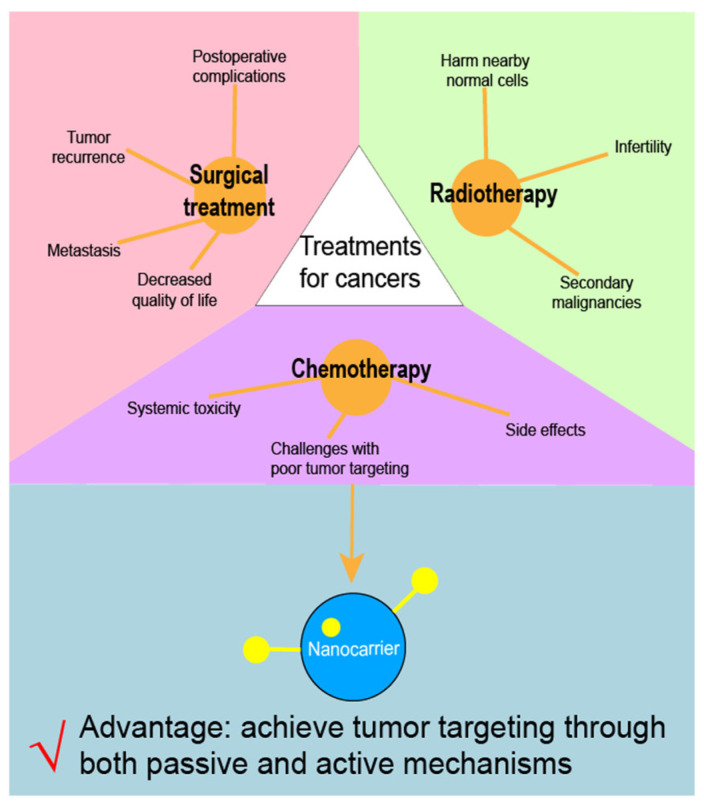
Limitations of three major kinds of cancer treatments.

**Figure 2 pharmaceuticals-18-00176-f002:**
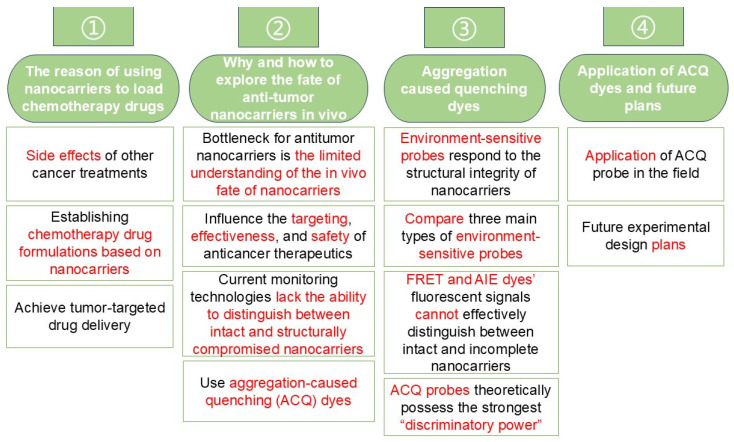
Mind map of this mini-review.

**Figure 3 pharmaceuticals-18-00176-f003:**
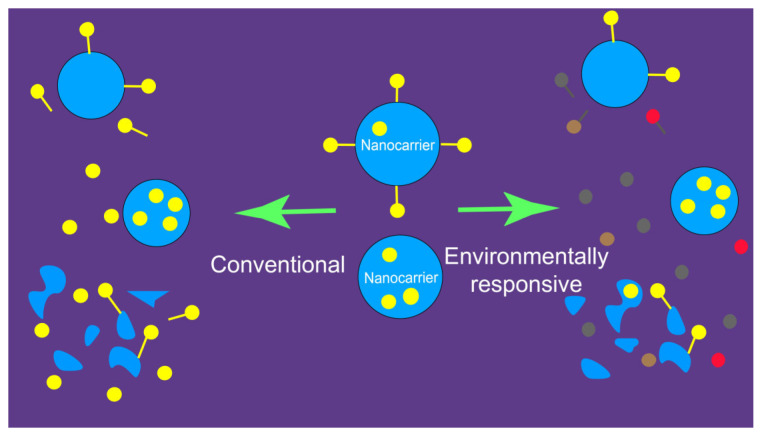
Disparity of fluorescence probe-labeled nanoparticulate drug delivery systems that were distinguishable and undistinguishable for the integrity. **Left panel:** Conventional probes; undistinguishable. **Right panel:** Environment-responsive probes, distinguishable [61].

**Figure 4 pharmaceuticals-18-00176-f004:**
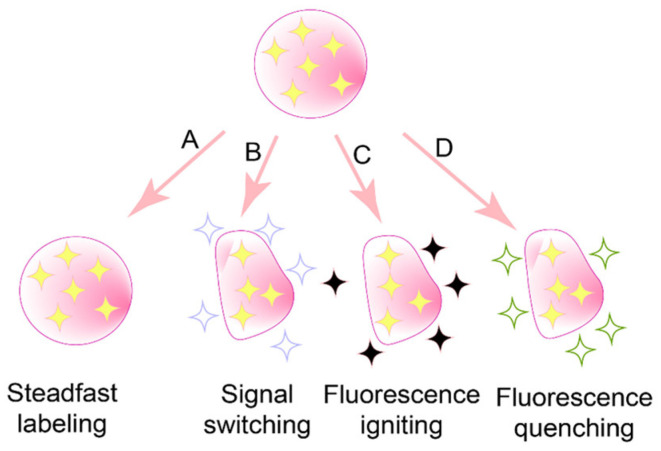
Available accesses to distinguish the integrity of probe-labeled nanoparticulate drug delivery systems [69]. Among them, (**A**) is not suitable because of poor biosafety, (**B**) is AIE probes, (**C**) is FRET probes, and (**D**) is ACQ probes.

**Figure 5 pharmaceuticals-18-00176-f005:**
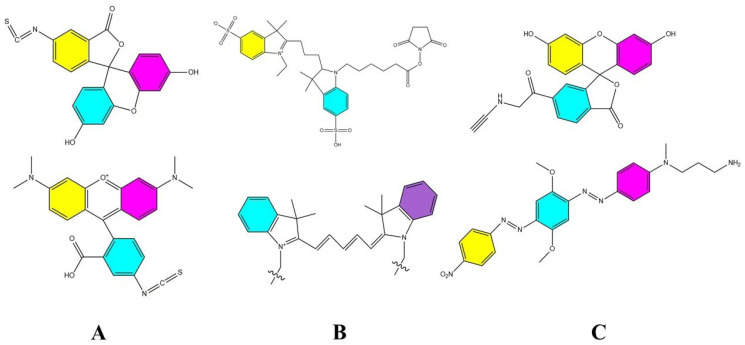
Common FRET probe pairs. (**A**) is FITC/TRITC [72], (**B**) is Cy3/Cy5, and (**C**) is FAM/BHQ-2 [73].

**Figure 6 pharmaceuticals-18-00176-f006:**
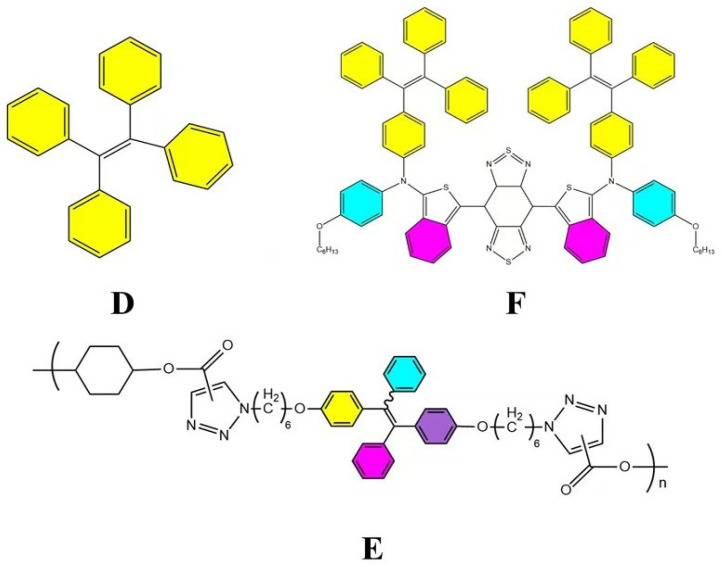
Common AIE probe pairs. (**D**) is TPE [80], (**E**) is PACT [80], and (**F**) is TPE-BT-BBTD [81].

**Figure 7 pharmaceuticals-18-00176-f007:**
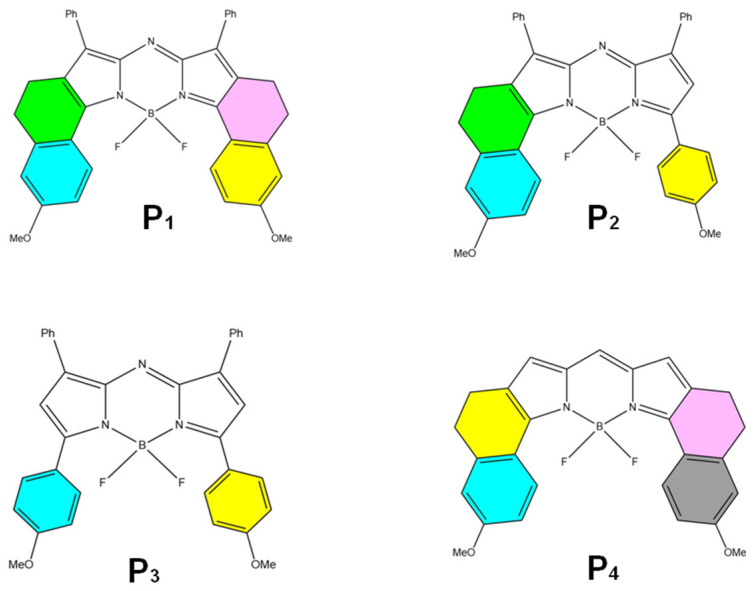
Common ACQ probes are the chemical structures of selected BODIPY and aza-BODIPY dyes (**P_1_** to **P_4_**) [68].

## Data Availability

No new data were created or analyzed in this study. Data sharing is not applicable to this article.

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
