# Peer review of "Aggregation-Caused Quenching Dyes as Potent Tools to Track the Integrity of Antitumor Nanocarriers: A Mini-Review"

_pharmaceuticals, 2025, doi:10.3390/ph18020176_

Round 1

Reviewer 1 Report

Comments and Suggestions for Authors

The manuscript entitled “Aggregation caused quenching dyes as potent tools to track the integrity of antitumor nanocarriers: A mini-review” was reviewed according to a review invitation by pharmaceuticals editorial office. My comments on how to improve the structure and content of the article are as follows.

1.       There are few typos (such as not respecting spaces between words and ref numbers) throughout the text that require careful proofreading.

2.      In the introduction, a little should be said about nanotechnology and its importance in cancer treatment, and then about nanocarriers.

3.      Figure 1 is too simple and do not provide useful info regarding to the manuscript title.

4.      The use of nanoparticles in chemotherapy is not limited to drug delivery. In many materials, nanoparticles are used to reduce the toxicity of chemotherapy drugs or to control drug release (https://doi.org/10.1016/j.heliyon.2023.e14024).

5.      The quality of Figure 2 is very poor and the text is generally unclear in the printed version.

6.       In Section 1, authors can highlight the methods introduced for designing the behavior of nanoparticles. For example, controlling drug release from nanocarriers by applying external stimuli (https://doi.org/10.1007/s12032-023-01991-1), or increasing the permeability of nanoparticles into cancer cells by applying external stimuli (https://doi.org/10.1007/s12032-022-01780-2), are examples of unconventional approaches for developing antitumor nanocarriers.

Author Response

Reviewer #1

The manuscript entitled “Aggregation caused quenching dyes as potent tools to track the integrity of antitumor nanocarriers: A mini-review” was reviewed according to a review invitation by pharmaceuticals editorial office. My comments on how to improve the structure and content of the article are as follows.

Response: Thanks for your positive comments. We have substantially improved the manuscript, and hope that the current version is acceptable.

Q1. There are few typos (such as not respecting spaces between words and ref numbers) throughout the text that require careful proofreading.

Response: Thanks for your valuable comments. We added Spaces between words and reference numbers. Please refer the new version of manuscript.

Q2. In the introduction, a little should be said about nanotechnology and its importance in cancer treatment, and then about nanocarriers.

Response: Thanks for your valuable comments. In the Introduction, we added detailed supplements of nanotechnology and its importance in cancer treatment, which is as follows:

The field of nanomedicine has witnessed unprecedented growth in the past few years, with continuous experimentation and testing leading to the development of modern nanostructures for the diagnosis and treatment of diseases, in particular, cancer1. Currently approved cancer nanomedicines are predominantly liposomal formulations and drug conjugates (proteins, polymers and/or antibodies), focusing on improving the pharmacokinetics and pharmacodynamics (PK/PD) of free drugs and utilizing passive targeting 2. The mechanisms of passive targeting is preferentially due to enhanced permeability and retention effects (EPR) 3. Targeting drug delivery is one of the main areas in which nanotechnology continues to fundamentally change the course of cancer treatment. Currently, nanostructures are being developed in two main ways with different modes: First, the nanostructures themselves can be used as both carriers and chemotherapeutic agents 4; Second, drugs can be absorbed directly into or dissolved within the nanoparticle framework, or covalently bound to the surface of the nanostructures 5. According to the above two modes, the nanostructures can be used to deliver drugs to the body as a carrier or as a chemotherapeutic agent per se1.

References:

  1. Mosleh-Shirazi S, Abbasi M, Moaddeli MR, et al. Nanotechnology Advances in the Detection and Treatment of Cancer: An Overview. Nanotheranostics 2022;6(4):400-23, 10.7150/ntno.74613.
  2. Kemp JA, Kwon YJ. Cancer nanotechnology: current status and perspectives. Nano Converg 2021;8(1):34, 10.1186/s40580-021-00282-7.
  3. Kenchegowda M, Rahamathulla M, Hani U, et al. Smart Nanocarriers as an Emerging Platform for Cancer Therapy: A Review. Molecules 2021;27(1), 10.3390/molecules27010146.
  4. Patra JK, Das G, Fraceto LF, et al. Nano based drug delivery systems: recent developments and future prospects. J Nanobiotechnology 2018;16(1):71, 10.1186/s12951-018-0392-8.
  5. Farzin L, Sheibani S, Moassesi ME, Shamsipur M. An overview of nanoscale radionuclides and radiolabeled nanomaterials commonly used for nuclear molecular imaging and therapeutic functions. J Biomed Mater Res A 2019;107(1):251-85, 10.1002/jbm.a.36550.

Q3. Figure 1 is too simple and do not provide useful info regarding to the manuscript title.

Response: Thanks for your valuable comments. We have modified Figure 1and the new one is in the word.

Q4. The use of nanoparticles in chemotherapy is not limited to drug delivery. In many materials, nanoparticles are used to reduce the toxicity of chemotherapy drugs or to control drug release (https://doi.org/10.1016/j.heliyon.2023.e14024).

Response: Thanks for your valuable comments. The literature you recommended was quite inspiring. We added the relevant introduction of nanoparticles used to reduce the toxicity of chemotherapy drugs or to control drug release, which is as follows:

The use of nanoparticles in chemotherapy is not limited to drug delivery. In many cases, nanoparticles were used to reduce the toxicity of chemotherapeutic drugs or to control drug release. Gold nanoparticles (AuNPs), for example, are ideally suited as diagnostic agents and for therapeutic use due to their unique properties, such as modulatable surface plasmon resonance (SPR), high surface reactivity, biocompatibility, and energy-absorbing capacity12. Studies have shown that AuNPs are toxic and quite difficult to manufacture. It was been demonstrated that the use of botanical phytochemicals in nanobiotechnology to encapsulate AuNPs can lower the toxicity and increase the feasibility for large-scale production3. Similarly, DNA nanoparticles (NPs) can also be used as carriers to wrap drugs for controlled release purposes. The carrier is suitable for wrapping gene and protein drugs and can protect the drugs from enzyme degradation. Specific targeting can also be achieved in vivo by changing the DNA structure, and the DNA nanoparticles (NPs) can also use its characteristics of specific changes with changes in temperature, light and pH to achieve controlled release. DNA nanoparticles (NPs) shows potential for treating multiple diseases4.

References:

  1. Amini SM, Kharrazi S, Jaafari MR. Radio frequency hyperthermia of cancerous cellswith gold nanoclusters: an in vitro investigation. Gold Bulletin 2017;50(1):43-50, 10.1007/s13404-016-0192-6.
  2. Koosha F, Farsangi ZJ, Samadian H, Amini SM. Mesoporous silica coated gold nanorods: a multifunctional theranostic platform for radiotherapy and X-ray imaging. Journal of Porous Materials 2021;28(6):1961-68, 10.1007/s10934-021-01137-6.
  3. Sharifiaghdam Z, Amini SM, Dalouchi F, Behrooz AB, Azizi Y. Apigenin-coated gold nanoparticles as a cardioprotective strategy against doxorubicin-induced cardiotoxicity in male rats via reducing apoptosis. Heliyon 2023;9(3):e14024, 10.1016/j.heliyon.2023.e14024.
  4. Kashani GK, Naghib SM, Soleymani S, Mozafari MR. A review of DNA nanoparticles-encapsulated drug/gene/protein for advanced controlled drug release: Current status and future perspective over emerging therapy approaches. International Journal of Biological Macromolecules 2024;268:131694, https://doi.org/10.1016/j.ijbiomac.2024.131694.

Q5. The quality of Figure 2 is very poor and the text is generally unclear in the printed version.

Response: Thanks for your valuable comments. We enhance the quality of Figure 2 and enlarge the text therein. The new figure is in the word

Q6. In Section 1, authors can highlight the methods introduced for designing the behavior of nanoparticles. For example, controlling drug release from nanocarriers by applying external stimuli (https://doi.org/10.1007/s12032-023-01991-1), or increasing the permeability of nanoparticles into cancer cells by applying external stimuli (https://doi.org/10.1007/s12032-022-01780-2), are examples of unconventional approaches for developing antitumor nanocarriers.

Response: Thanks for your valuable comments. The literature you recommended was quite inspiring. We agree that the designing methods of nanoparticles are very important, and therefore decide to add a new relevant section, as it seemed less adherent to be incorporated into Section 1. The new Section 1 about controlling drug release from nanocarriers by applying external stimuli (Section 1.1) and increasing the permeability of nanoparticles into cancer cells by applying external stimuli (Section 1.2) is attached as follows:

1 Major strategies for developing antitumor nanocarriers

Nowadays, drug incorporation methods for nanoparticles have become abundant and mature. Traditional incorporation methods include: using liposomal nanoparticles to encapsulate drugs1, which selectively bind to target cells and release the drugs; using magnetic nanoparticle targeted delivery systems combined with the action of an external magnetic field to precisely deliver drugs to specific organs and tissues2; encapsulating nucleic acid drugs in lipid nanoparticles, followed by surface modification with specific ligands or antibodies, the lipid nanoparticles are able to target cells or tissues and release the drugs3. We can integrate these methods into two dominant domains, viz. to control drug release and improve cellular uptake. Hereby, this mini-review will briefly summarize the representative cases upon controlling drug release and increasing the permeability into cancer cells.

1.1. Controlling drug release

Controlling drug release of antitumor nanocarriers can be achieved by applying external stimuli and taking advantages of nanocarrier nature, which is introduced below.

1.1.1. Applying external stimuli

Distinguished from unresponsive anti-tumor nanocarrier designs, we believe that the design of applying external stimuli to control drug release is novel and effective. In contrast to endogenous stimuli in biological environments, the responsiveness of nanoparticles to exogenous stimuli allows for remote control of the action at specific points in space and time. Once acting externally on the target tissue, the exogenous stimulus can be easily controlled after loading a drug to a specific nanoparticle. This ensures that rationally designed nanoparticles can be activated at the demanded time and site. It was evidenced that the design approach of using external stimuli to facilitate drug release is feasible 4. For example, 2-nm nanoparticulate gold clusters with good heating effect under radiofrequency electric field (RF-EF) irradiation have been used for in vitro thermotherapy of cancer cells. The researchers prepared doxorubicin liposomes and exposed the prepared formulations to RF-EF. The experimental results demonstrated that 20.2 ± 2.1% of the drug was released and the IC50 value of colorectal cancer cells decreased twofold. The X-ray attenuation efficiency of liposomal gold clusters was superior to that of commercial iodohexitol and free gold clusters at different concentrations. Finally, treatment of cancer cells with gold clusters significantly reduced cell survival under cobalt-60 beam irradiation 5.

Typical features of hypoxic cells, such as low oxygen levels and highly bioreductive environments, may provide stimulus-responsive drug release to aid in tumor-specific chemotherapy, radiotherapy, photodynamic therapy, and acoustic kinetic therapy. Experimental results demonstrated the successful development of hypoxia-responsive nanocarriers (Tumor microenvironment (TME) receptive and modulating therapeutic nano drug delivery systems (DDS)) that could be used for drug delivery to heterogeneous tumors. This approach of targeting hypoxic tumor habitats was expected to overcome difficulties arising from tumor heterogeneity and could be used to design diagnostic and therapeutic nanocarriers to target various types of solid tumors 6.

1.1.2. Taking advantages of nanocarrier nature

Triggering of drug release can be designed directly by utilizing the inherent properties of the nanocarrier material; however, it should be noted that stimulus are still needed in this scenario. Confronting the challenge that nanomaterials suffer from insufficient delivery of therapeutic drugs at the target site, in recent years, researchers have developed stimuli-responsive nanocarriers 7-11, which change their properties when subjected to various stimuli, in order to deliver drugs and genes to the target site in a controlled and adequate manner. The classical examples include Stimuli-responsive polymers for antimicrobial therapy12, a DNA delivery nanocarrier called Pluronic-PEI-SS synthesized by conjugating reducible disulfide-linked PEI (PEI-SS) with Pluronic was fabricated13, Enzyme-Responsive Liposomes14, Stimuli-responsive nanocarriers15.

Of note, stimuli-responsive nanocarriers usually contain various environmentally sensitive features in their structures, so that the loaded therapeutic drugs can be released in response to various environmental factors, such as endogenous stimuli (e.g., temperature, pH, redox potential, enzymes, etc.) and exogenous stimuli (e.g., electromagnetism, light, radiation, ultrasound, etc.) 16. By this means, versatile nanocarriers can be developed.

1.2. Increasing the permeability into cancer cells

Similar to controlling drug release, increasing the permeability into cancer cells of antitumor nanocarriers can be achieved by applying external stimuli and taking advantages of nanocarrier nature as well.

1.2.1. Applying external stimuli

Increased accumulation of nanoparticles in tumors using the EPR effect is known as a passive targeting approach 17, which is widely recognized. However, relying solely on EPR effect may be insufficient for tumoral accumulation, and external stimuli should be introduced. In a study using folate-functionalized gold magnetic core-shell nanostructures for the treatment of HPV (Human Papillomavirus)-positive oral cancer, HPV-positive oral cancer cells were subjected to nanoparticles, electric fields, and radiation treatments. Tumor cell viability and apoptosis rates were determined, and cellular uptake of the nanoparticles was also determined by Inductively Coupled Plasma-Optical Emission Spectrometer (ICP-OES) analysis. It was found that none of the treatments alone resulted in significant cancer cell death. Combined treatments increased the mortality of cancer cells and increased the proportion of apoptotic cells among them. It was also observed that the electric field enhanced the uptake of the nanoparticles by the cancer cells, and it could be concluded that the combination of folic acid-functionalized nanoparticles and electroporation opened up a new avenue for improving the efficacy of radiation therapy for human papillomavirus-positive cancers. This finding is integral to the application of external stimuli to increase the permeability of the nanoparticles into the cancer cells 18.

1.2.2. Taking advantages of nanocarrier nature

Novel strategies such as transformable nanocarriers, transcellular delivery of peptide-modified nanocarriers, and bio-inspired carriers have recently emerged as a new generation of drug carriers with uptake enhancement possibilities17. For example, polymeric nanoparticles (like Adriamycin-encapsulated nanoliposomes with polyethylene glycol (PEG) coating enhance the circulating half-life of the drug required for the treatment of cancer (Kaposi's sarcoma, recurrent ovarian cancer, etc.)19. have a small size range and variable shape. Their small size allows them to penetrate and be taken up by cells, thereby increasing the accumulation of the drug at the targeted tumor site. Several methods can be used to incorporate drugs into polymeric nanoparticles, such as dissolution, precipitation, adsorption or attachment 20-23. Albumin nanoparticle carrier systems were designed to take advantage of the presence of different drug binding sites in the albumin molecule and the fact that a large number of drugs can be doped into the particle matrix24. Albumin nanoparticles are easy to prepare, smaller in size, biodegradable, non-toxic, harmless, easy to metabolize and are very suitable natural polymers25.

References:

  1. Guerrero G, Mutin PH, Vioux A. Mixed Nonhydrolytic/Hydrolytic Sol−Gel Routes to Novel Metal Oxide/Phosphonate Hybrids. Chemistry of Materials 2000;12(5):1268-72, 10.1021/cm991125+.
  2. kianfar E. Magnetic Nanoparticles in Targeted Drug Delivery: a Review. Journal of Superconductivity and Novel Magnetism 2021;34(7):1709-35, 10.1007/s10948-021-05932-9.
  3. Samaridou E, Heyes J, Lutwyche P. Lipid nanoparticles for nucleic acid delivery: Current perspectives. Advanced Drug Delivery Reviews 2020;154-155, 10.1016/j.addr.2020.06.002.
  4. Pruchyathamkorn J, Yang M, Amin HMA, Batchelor-McAuley C, Compton RG. Imaging Electrode Heterogeneity Using Chemically Confined Fluorescence Electrochemical Microscopy. The Journal of Physical Chemistry Letters 2017;8(24):6124-27, 10.1021/acs.jpclett.7b02925.
  5. Amini SM, Rezayat SM, Dinarvand R, Kharrazi S, Jaafari MR. Gold cluster encapsulated liposomes: theranostic agent with stimulus triggered release capability. Med Oncol 2023;40(5):126, 10.1007/s12032-023-01991-1.
  6. Kumari R, Sunil D, Ningthoujam RS. Hypoxia-responsive nanoparticle based drug delivery systems in cancer therapy: An up-to-date review. J Control Release 2020;319:135-56, 10.1016/j.jconrel.2019.12.041.
  7. Onaca O, Enea R, Hughes DW, Meier W. Stimuli-responsive polymersomes as nanocarriers for drug and gene delivery. Macromol Biosci 2009;9(2):129-39, 10.1002/mabi.200800248.
  8. Rao NV, Ko H, Lee J, Park JH. Recent Progress and Advances in Stimuli-Responsive Polymers for Cancer Therapy. Front Bioeng Biotechnol 2018;6:110, 10.3389/fbioe.2018.00110.
  9. Nakayama M, Akimoto J, Okano T. Polymeric micelles with stimuli-triggering systems for advanced cancer drug targeting. J Drug Target 2014;22(7):584-99, 10.3109/1061186x.2014.936872.
  10. Hu YW, Du YZ, Liu N, et al. Selective redox-responsive drug release in tumor cells mediated by chitosan based glycolipid-like nanocarrier. J Control Release 2015;206:91-100, 10.1016/j.jconrel.2015.03.018.
  11. Mura S, Nicolas J, Couvreur P. Stimuli-responsive nanocarriers for drug delivery. Nature Materials 2013;12(11):991-1003, 10.1038/nmat3776.
  12. Alvarez-Lorenzo C, Garcia-Gonzalez CA, Bucio E, Concheiro A. Stimuli-responsive polymers for antimicrobial therapy: drug targeting, contact-killing surfaces and competitive release. Expert Opinion on Drug Delivery 2016;13(8):1109-19, 10.1080/17425247.2016.1178719.
  13. Zhang Y, Yu J, Kahkoska AR, Wang J, Buse JB, Gu Z. Advances in transdermal insulin delivery. Advanced Drug Delivery Reviews 2019;139:51-70,
  14. Fouladi F, Steffen KJ, Mallik S. Enzyme-Responsive Liposomes for the Delivery of Anticancer Drugs. Bioconjugate Chemistry 2017;28(4):857-68,
  15. Tayo LL. Stimuli-responsive nanocarriers for intracellular delivery. Biophysical Reviews 2017;9(6):931-40, 10.1007/s12551-017-0341-z.
  16. Majumder J, Minko T. Multifunctional and stimuli-responsive nanocarriers for targeted therapeutic delivery. Expert Opin Drug Deliv 2021;18(2):205-27, 10.1080/17425247.2021.1828339.
  17. Souri M, Soltani M, Moradi Kashkooli F, Kiani Shahvandi M. Engineered strategies to enhance tumor penetration of drug-loaded nanoparticles. Journal of Controlled Release 2022;341:227-46,
  18. Ahmadi Kamalabadi M, Neshastehriz A, Ghaznavi H, Amini SM. Folate functionalized gold-coated magnetic nanoparticles effect in combined electroporation and radiation treatment of HPV-positive oropharyngeal cancer. Med Oncol 2022;39(12):196, 10.1007/s12032-022-01780-2.
  19. Barenholz Y. Doxil® — The first FDA-approved nano-drug: Lessons learned. Journal of Controlled Release 2012;160(2):117-34,
  20. Kumari A, Yadav SK, Yadav SC. Biodegradable polymeric nanoparticles based drug delivery systems. Colloids and Surfaces B: Biointerfaces 2010;75(1):1-18,
  21. Pinto Reis C, Neufeld RJ, Ribeiro AJ, Veiga F. Nanoencapsulation I. Methods for preparation of drug-loaded polymeric nanoparticles. Nanomedicine: Nanotechnology, Biology and Medicine 2006;2(1):8-21,.
  22. Tang L, Azzi J, Kwon M, et al. Immunosuppressive Activity of Size-Controlled PEG-PLGA Nanoparticles Containing Encapsulated Cyclosporine A. J Transplant 2012;2012:896141, 10.1155/2012/896141.
  23. Venditti I. Morphologies and functionalities of polymeric nanocarriers as chemical tools for drug delivery: A review. Journal of King Saud University - Science 2019;31(3):398-411,
  24. Leng DD, Han WJ, Rui Y, Dai Y, Xia YF. In vivo disposition and metabolism of madecassoside, a major bioactive constituent in Centella asiatica (L.) Urb. J Ethnopharmacol 2013;150(2):601-8, 10.1016/j.jep.2013.09.004.
  25. Elzoghby AO, Samy WM, Elgindy NA. Albumin-based nanoparticles as potential controlled release drug delivery systems. Journal of controlled release : official journal of the Controlled Release Society 2012;157(2):168-82, 10.1016/j.jconrel.2011.07.031.

Reviewer 2 Report

Comments and Suggestions for Authors

This minireview reports on the in vivo fate of nanocarriers affects the therapeutic efficacy of chemotherapeutic nanoformulations, including factors such as nanocarrier stability and tumor-targeting capacity. The aggregation-caused quenching (ACQ) dyes—characterized by their sensitive water-quenching properties and easily detectable “on-off” switching behavior—to track the integrity of antitumor nanocarriers in vivo following administration. However, some points have to be addressed before publication:

1.       Why the abstract is written in a style of “background, methods, results,..”. This is a review and not a research paper. Rewrite these parts.

2.       The review should include a discussion of some case studies and examples of the nanocarriers and discuss their performances.

3.       The use of fluorescent probes to visualize the interaction regions should be strengthened by citing the following work: Imaging electrode heterogeneity using chemically confined fluorescence electrochemical microscopy.

4.       Sentence from L56 on the development of drug delivery systems should be supported by this reference doi.org/10.1016/j.ijbiomac.2024.138156

5.       The conclusion should also highlight the challenges and limitations in this field.

Author Response

Reviewer #2

This minireview reports on the in vivo fate of nanocarriers affects the therapeutic efficacy of chemotherapeutic nanoformulations, including factors such as nanocarrier stability and tumor-targeting capacity. The aggregation-caused quenching (ACQ) dyes—characterized by their sensitive water-quenching properties and easily detectable “on-off” switching behavior—to track the integrity of antitumor nanocarriers in vivo following administration. However, some points have to be addressed before publication:

Response: Thanks for your positive comments. We have substantially improved the manuscript, and hope that the current version is acceptable.

Q1. Why the abstract is written in a style of “background, methods, results,..”. This is a review and not a research paper. Rewrite these parts.

Response: Thanks for your valuable comments. We were sorry for using the misleading style. The structural words like “background, methods, results,..” in the Abstract were deleted in the new version. The new abstract is as follows:

Cancer has become one of the major causes of death worldwide. Chemotherapy remains a cornerstone of cancer treatment. To enhance the tumor-targeting efficiency of chemotherapy agents, pharmaceutical scientists have developed nanocarriers. However, the in vivo structural integrity and dynamic changes of nanocarriers after administration are not well understood, which may significantly impact their tumor-targeting abilities. In this paper, we propose the use of environmentally responsive fluorescent probes to track the integrity of antitumor nanocarriers. We compare three main types of dyes: fluorescence resonance energy transfer (FRET) dyes, aggregation-induced emission (AIE) dyes, and aggregation-caused quenching (ACQ) dyes. Among them, ACQ dyes, possessing sensitive water-quenching properties and easily detected “on-off” switching behavior, are regarded as the most promising choice. We believe that ACQ dyes are suitable for investigating the in vivo fate of antitumor nanocarriers and can aid in designing improved nanoformulations for chemotherapy agents.

Q2. The review should include a discussion of some case studies and examples of the nanocarriers and discuss their performances.

Response: Thanks for your valuable comments. We now added a new Section 1 Major strategies for developing antitumor nanocarriers in the revised manuscript, and discussed some case studies and examples of the nanocarriers and their performances, which is as follows:

1 Major strategies for developing antitumor nanocarriers

Nowadays, drug incorporation methods for nanoparticles have become abundant and mature. Traditional incorporation methods include: using liposomal nanoparticles to encapsulate drugs1, which selectively bind to target cells and release the drugs; using magnetic nanoparticle targeted delivery systems combined with the action of an external magnetic field to precisely deliver drugs to specific organs and tissues2; encapsulating nucleic acid drugs in lipid nanoparticles, followed by surface modification with specific ligands or antibodies, the lipid nanoparticles are able to target cells or tissues and release the drugs3. We can integrate these methods into two dominant domains, viz. to control drug release and improve cellular uptake. Hereby, this mini-review will briefly summarize the representative cases upon controlling drug release and increasing the permeability into cancer cells.

1.1. Controlling drug release

Controlling drug release of antitumor nanocarriers can be achieved by applying external stimuli and taking advantages of nanocarrier nature, which is introduced below.

1.1.1. Applying external stimuli

Distinguished from unresponsive anti-tumor nanocarrier designs, we believe that the design of applying external stimuli to control drug release is novel and effective. In contrast to endogenous stimuli in biological environments, the responsiveness of nanoparticles to exogenous stimuli allows for remote control of the action at specific points in space and time. Once acting externally on the target tissue, the exogenous stimulus can be easily controlled after loading a drug to a specific nanoparticle. This ensures that rationally designed nanoparticles can be activated at the demanded time and site. It was evidenced that the design approach of using external stimuli to facilitate drug release is feasible 4. For example, 2-nm nanoparticulate gold clusters with good heating effect under radiofrequency electric field (RF-EF) irradiation have been used for in vitro thermotherapy of cancer cells. The researchers prepared doxorubicin liposomes and exposed the prepared formulations to RF-EF. The experimental results demonstrated that 20.2 ± 2.1% of the drug was released and the IC50 value of colorectal cancer cells decreased twofold. The X-ray attenuation efficiency of liposomal gold clusters was superior to that of commercial iodohexitol and free gold clusters at different concentrations. Finally, treatment of cancer cells with gold clusters significantly reduced cell survival under cobalt-60 beam irradiation 5.

Typical features of hypoxic cells, such as low oxygen levels and highly bioreductive environments, may provide stimulus-responsive drug release to aid in tumor-specific chemotherapy, radiotherapy, photodynamic therapy, and acoustic kinetic therapy. Experimental results demonstrated the successful development of hypoxia-responsive nanocarriers (Tumor microenvironment (TME) receptive and modulating therapeutic nano drug delivery systems (DDS)) that could be used for drug delivery to heterogeneous tumors. This approach of targeting hypoxic tumor habitats was expected to overcome difficulties arising from tumor heterogeneity and could be used to design diagnostic and therapeutic nanocarriers to target various types of solid tumors 6.

1.1.2. Taking advantages of nanocarrier nature

Triggering of drug release can be designed directly by utilizing the inherent properties of the nanocarrier material; however, it should be noted that stimulus are still needed in this scenario. Confronting the challenge that nanomaterials suffer from insufficient delivery of therapeutic drugs at the target site, in recent years, researchers have developed stimuli-responsive nanocarriers 7-11, which change their properties when subjected to various stimuli, in order to deliver drugs and genes to the target site in a controlled and adequate manner. The classical examples include Stimuli-responsive polymers for antimicrobial therapy12, a DNA delivery nanocarrier called Pluronic-PEI-SS synthesized by conjugating reducible disulfide-linked PEI (PEI-SS) with Pluronic was fabricated13, Enzyme-Responsive Liposomes14, Stimuli-responsive nanocarriers15.

Of note, stimuli-responsive nanocarriers usually contain various environmentally sensitive features in their structures, so that the loaded therapeutic drugs can be released in response to various environmental factors, such as endogenous stimuli (e.g., temperature, pH, redox potential, enzymes, etc.) and exogenous stimuli (e.g., electromagnetism, light, radiation, ultrasound, etc.) 16. By this means, versatile nanocarriers can be developed.

1.2. Increasing the permeability into cancer cells

Similar to controlling drug release, increasing the permeability into cancer cells of antitumor nanocarriers can be achieved by applying external stimuli and taking advantages of nanocarrier nature as well.

1.2.1. Applying external stimuli

Increased accumulation of nanoparticles in tumors using the EPR effect is known as a passive targeting approach 17, which is widely recognized. However, relying solely on EPR effect may be insufficient for tumoral accumulation, and external stimuli should be introduced. In a study using folate-functionalized gold magnetic core-shell nanostructures for the treatment of HPV (Human Papillomavirus)-positive oral cancer, HPV-positive oral cancer cells were subjected to nanoparticles, electric fields, and radiation treatments. Tumor cell viability and apoptosis rates were determined, and cellular uptake of the nanoparticles was also determined by Inductively Coupled Plasma-Optical Emission Spectrometer (ICP-OES) analysis. It was found that none of the treatments alone resulted in significant cancer cell death. Combined treatments increased the mortality of cancer cells and increased the proportion of apoptotic cells among them. It was also observed that the electric field enhanced the uptake of the nanoparticles by the cancer cells, and it could be concluded that the combination of folic acid-functionalized nanoparticles and electroporation opened up a new avenue for improving the efficacy of radiation therapy for human papillomavirus-positive cancers. This finding is integral to the application of external stimuli to increase the permeability of the nanoparticles into the cancer cells 18.

1.2.2. Taking advantages of nanocarrier nature

Novel strategies such as transformable nanocarriers, transcellular delivery of peptide-modified nanocarriers, and bio-inspired carriers have recently emerged as a new generation of drug carriers with uptake enhancement possibilities17. For example, polymeric nanoparticles (like Adriamycin-encapsulated nanoliposomes with polyethylene glycol (PEG) coating enhance the circulating half-life of the drug required for the treatment of cancer (Kaposi's sarcoma, recurrent ovarian cancer, etc.)19. have a small size range and variable shape. Their small size allows them to penetrate and be taken up by cells, thereby increasing the accumulation of the drug at the targeted tumor site. Several methods can be used to incorporate drugs into polymeric nanoparticles, such as dissolution, precipitation, adsorption or attachment 20-23. Albumin nanoparticle carrier systems were designed to take advantage of the presence of different drug binding sites in the albumin molecule and the fact that a large number of drugs can be doped into the particle matrix24. Albumin nanoparticles are easy to prepare, smaller in size, biodegradable, non-toxic, harmless, easy to metabolize and are very suitable natural polymers25.

References:

  1. Guerrero G, Mutin PH, Vioux A. Mixed Nonhydrolytic/Hydrolytic Sol−Gel Routes to Novel Metal Oxide/Phosphonate Hybrids. Chemistry of Materials 2000;12(5):1268-72, 10.1021/cm991125+.
  2. kianfar E. Magnetic Nanoparticles in Targeted Drug Delivery: a Review. Journal of Superconductivity and Novel Magnetism 2021;34(7):1709-35, 10.1007/s10948-021-05932-9.
  3. Samaridou E, Heyes J, Lutwyche P. Lipid nanoparticles for nucleic acid delivery: Current perspectives. Advanced Drug Delivery Reviews 2020;154-155, 10.1016/j.addr.2020.06.002.
  4. Pruchyathamkorn J, Yang M, Amin HMA, Batchelor-McAuley C, Compton RG. Imaging Electrode Heterogeneity Using Chemically Confined Fluorescence Electrochemical Microscopy. The Journal of Physical Chemistry Letters 2017;8(24):6124-27, 10.1021/acs.jpclett.7b02925.
  5. Amini SM, Rezayat SM, Dinarvand R, Kharrazi S, Jaafari MR. Gold cluster encapsulated liposomes: theranostic agent with stimulus triggered release capability. Med Oncol 2023;40(5):126, 10.1007/s12032-023-01991-1.
  6. Kumari R, Sunil D, Ningthoujam RS. Hypoxia-responsive nanoparticle based drug delivery systems in cancer therapy: An up-to-date review. J Control Release 2020;319:135-56, 10.1016/j.jconrel.2019.12.041.
  7. Onaca O, Enea R, Hughes DW, Meier W. Stimuli-responsive polymersomes as nanocarriers for drug and gene delivery. Macromol Biosci 2009;9(2):129-39, 10.1002/mabi.200800248.
  8. Rao NV, Ko H, Lee J, Park JH. Recent Progress and Advances in Stimuli-Responsive Polymers for Cancer Therapy. Front Bioeng Biotechnol 2018;6:110, 10.3389/fbioe.2018.00110.
  9. Nakayama M, Akimoto J, Okano T. Polymeric micelles with stimuli-triggering systems for advanced cancer drug targeting. J Drug Target 2014;22(7):584-99, 10.3109/1061186x.2014.936872.
  10. Hu YW, Du YZ, Liu N, et al. Selective redox-responsive drug release in tumor cells mediated by chitosan based glycolipid-like nanocarrier. J Control Release 2015;206:91-100, 10.1016/j.jconrel.2015.03.018.
  11. Mura S, Nicolas J, Couvreur P. Stimuli-responsive nanocarriers for drug delivery. Nature Materials 2013;12(11):991-1003, 10.1038/nmat3776.
  12. Alvarez-Lorenzo C, Garcia-Gonzalez CA, Bucio E, Concheiro A. Stimuli-responsive polymers for antimicrobial therapy: drug targeting, contact-killing surfaces and competitive release. Expert Opinion on Drug Delivery 2016;13(8):1109-19, 10.1080/17425247.2016.1178719.
  13. Zhang Y, Yu J, Kahkoska AR, Wang J, Buse JB, Gu Z. Advances in transdermal insulin delivery. Advanced Drug Delivery Reviews 2019;139:51-70, https://doi.org/10.1016/j.addr.2018.12.006.
  14. Fouladi F, Steffen KJ, Mallik S. Enzyme-Responsive Liposomes for the Delivery of Anticancer Drugs. Bioconjugate Chemistry 2017;28(4):857-68, https://doi.org/10.1021/acs.bioconjchem.6b00736.
  15. Tayo LL. Stimuli-responsive nanocarriers for intracellular delivery. Biophysical Reviews 2017;9(6):931-40, 10.1007/s12551-017-0341-z.
  16. Majumder J, Minko T. Multifunctional and stimuli-responsive nanocarriers for targeted therapeutic delivery. Expert Opin Drug Deliv 2021;18(2):205-27, 10.1080/17425247.2021.1828339.
  17. Souri M, Soltani M, Moradi Kashkooli F, Kiani Shahvandi M. Engineered strategies to enhance tumor penetration of drug-loaded nanoparticles. Journal of Controlled Release 2022;341:227-46, https://doi.org/10.1016/j.jconrel.2021.11.024.
  18. Ahmadi Kamalabadi M, Neshastehriz A, Ghaznavi H, Amini SM. Folate functionalized gold-coated magnetic nanoparticles effect in combined electroporation and radiation treatment of HPV-positive oropharyngeal cancer. Med Oncol 2022;39(12):196, 10.1007/s12032-022-01780-2.
  19. Barenholz Y. Doxil® — The first FDA-approved nano-drug: Lessons learned. Journal of Controlled Release 2012;160(2):117-34, https://doi.org/10.1016/j.jconrel.2012.03.020.
  20. Kumari A, Yadav SK, Yadav SC. Biodegradable polymeric nanoparticles based drug delivery systems. Colloids and Surfaces B: Biointerfaces 2010;75(1):1-18, https://doi.org/10.1016/j.colsurfb.2009.09.001.
  21. Pinto Reis C, Neufeld RJ, Ribeiro AJ, Veiga F. Nanoencapsulation I. Methods for preparation of drug-loaded polymeric nanoparticles. Nanomedicine: Nanotechnology, Biology and Medicine 2006;2(1):8-21, https://doi.org/10.1016/j.nano.2005.12.003.
  22. Tang L, Azzi J, Kwon M, et al. Immunosuppressive Activity of Size-Controlled PEG-PLGA Nanoparticles Containing Encapsulated Cyclosporine A. J Transplant 2012;2012:896141, 10.1155/2012/896141.
  23. Venditti I. Morphologies and functionalities of polymeric nanocarriers as chemical tools for drug delivery: A review. Journal of King Saud University - Science 2019;31(3):398-411, https://doi.org/10.1016/j.jksus.2017.10.004.
  24. Leng DD, Han WJ, Rui Y, Dai Y, Xia YF. In vivo disposition and metabolism of madecassoside, a major bioactive constituent in Centella asiatica (L.) Urb. J Ethnopharmacol 2013;150(2):601-8, 10.1016/j.jep.2013.09.004.
  25. Elzoghby AO, Samy WM, Elgindy NA. Albumin-based nanoparticles as potential controlled release drug delivery systems. Journal of controlled release : official journal of the Controlled Release Society 2012;157(2):168-82, 10.1016/j.jconrel.2011.07.031.

Q3. The use of fluorescent probes to visualize the interaction regions should be strengthened by citing the following work: Imaging electrode heterogeneity using chemically confined fluorescence electrochemical microscopy.

Response: Thanks for your valuable comments. The literature you recommended was quite inspiring. We added “In this case, imaging electrode heterogeneity using chemically constrained fluorescence electrochemical microscopy enables visualization of the interaction regions of fluorescent probes1.” after the second paragraph of section 3.

References:

  1. Pruchyathamkorn J, Yang M, Amin HMA, Batchelor-McAuley C, Compton RG. Imaging Electrode Heterogeneity Using Chemically Confined Fluorescence Electrochemical Microscopy. The Journal of Physical Chemistry Letters 2017;8(24):6124-27, 10.1021/acs.jpclett.7b02925.

Q4. Sentence from L56 on the development of drug delivery systems should be supported by this reference doi.org/10.1016/j.ijbiomac.2024.13815626

Response: Thanks for your valuable comments. The literature you recommended was quite inspiring. The literature is referenced in the previous L56.

References:

  1. Feroze F, Sher M, Hussain MA, et al. Gastro retentive floating drug delivery system of levofloxacin based on Aloe vera hydrogel: In vitro and in vivo assays. International Journal of Biological Macromolecules 2025;284:138156, https://doi.org/10.1016/j.ijbiomac.2024.138156.

Q5. The conclusion should also highlight the challenges and limitations in this field.

Response: Thanks for your valuable comments. We changed the section title from “Conclusion” to “Concluding remarks”, and highlight the challenges and limitations in this field. We added “Although the future development of fluorescent probes is very promising, in the real-world process of application, the properties of fluorescent probes may be affected by the physiological environment and other various reasons. For quick example, the reactive oxygen monitoring probe will be affected by various chemical interactions and may not be specific1; in the process of nitrite detection in the body, the fluorescent probe may become not sensitive enough2. These shortcomings still encourage us to carry out continuous studies and find suitable strategies to circumvent them.” as a new paragraph after the first paragraph of section 4.

References:

  1. Winterbourn CC. The challenges of using fluorescent probes to detect and quantify specific reactive oxygen species in living cells. Biochim Biophys Acta 2014;1840(2):730-8, 10.1016/j.bbagen.2013.05.004.
  2. Zhang Q, Wang Y, Song A, Yang X, Yin D, Shen L. Advancements in fluorescent probes for nitrite sensing: A review. Journal of Molecular Structure 2023,

Reviewer 3 Report

Comments and Suggestions for Authors

Aggregation caused quenching dyes as potent tools to track the integrity of antitumor nanocarriers: A mini-review

The present mini-review has as main goal the subject of drug delivery systems used in different cancer therapies. This is a field of great interest nowadays, while different types of cancer are the major cause of death worldwide. Therefore, a continuous research is performed to develop better drug delivery systems, as for instance nanocarriers, described in the current manuscript.

The mini-review is in general well written, the Abstract is concise and clear and the References are relevant to the field. At the key-words „antitumor” and „drug delivery” could be added. The ithenticate percentage of similarities is very low, considering the fact that it represents actually a review which show data and results already published in the literature. All the abbreviations are explained immediately when they appear for the first time and this will help thre readers. The English is fine trough the whole manuscript.

The main options currently available in general for the treatments of different cancers, are described in details: radiotherapy, anti-cancer medication and surgery. But, for a review in such a field of great interest, with a lot of recent bibliographic sources, I would expect more citations. The references list is rather short. The authors should add more references. For exampe, I did not found mentioned the osteo cancers, a type of tumor with an increasing incidence, especially for the younger population. In the literature there are several examples which proved that some organic-inorganic hybrids phosphorus containing could have a positive effect on malignant cells of human osteo-sarcoma, as for example the work of Khaled et al. https://doi.org/10.3390/cimb46050290

Such hybrid compounds containing phosphorus could be biocompatible and bioactive materials (in addition they could also contain zirconium or titanium, for example, already used in different applications in medicine), and could be synthesized by using the sol-gel method as performed in the work published by Mehring et al. https://doi.org/10.1023/A:1020797520620 , Guerrero et al. https://doi.org/10.1021/cm991125+ , Sanchez et al. https://doi.org/10.1023/A:1008753919925 , Mutin et al. https://doi.org/10.2109/jcersj2.123.709 and those are just few examples. The authors could cite them and could also find more references, especially from the last decade, relevant to the field of the present review. Moreover, the sol-gel method could be mentioned in addition to other methods reported in the review, while also the hydrogels are mentioned in the citations of the work of Zhang et al. [8 and 23].

And, as recommendations also here, please cite the references according to the guide for authors. Therefore, the names of the first 3 authors should be mentioned when a reference is cited as bibliographic source, before to mention „et al.”. So, "Zhang et al." is not correctly cited; as a consequence, you need to add the other autors and only after the third author to mention „et al.”. There are many mistakes in the references list according to the journal citations rules for authors. Please check all the list, because most of the citations mention only the first author. But, for instance, in the case of 5, 10, 11, 19, 22, 43, 44, 45, 48, 49 two authors are mentioned, and in the case of 16, 18, 36, 41, 47 three authors are mentioned. All of those are just some examples. Moreover, in some cases, the author’s initial letter of the first name is added after the family name, and in other cases before the family name (for example: Warde, U. and N.J.J.o.F. Sekar; Mukherjee, S. and P.J.C.C. Thilagar; Zhao, W. and E.M.J.A.C. Carreira; Zhao, W. and E.M.J.C.A.E.J. Carreira; Wu, L. and K.J.C.c. Burgess). The same style should be used in all cases, as recommended by the journal. Nevetheless, for the authors with more initial letters (therefore, more names) only cappital letters should be used (not N.J.J.o.F. or K.J.C.c., for instance; correctly will be N.J.J.O.F. and K.J.C.C. in those cases). Also you need to check if the DOI number is needed to be included in all the citations from the references list.

The methods and the results are described generally in details. Still, because the main subject is related to nanocarriers as drug delivery systems and to the aggregation (as mentioned both already from the title), it would be interesting and useful for the readers to add a discussion about the aggregation in terms of colloids. The aggregatoin could occur due to DLVO and non-DLVO forces, as electrostatic forces, van der Waals, structural forces, hydration forces, micellar interactions, polymer networks if it is the case and so on. Those are theories of forces in colloidal chemistry. Are those aggregations, colloidal dispersions? If so, all of these should be discussed, in terms of charge, particles size, hydrophobic / hidrophilic character, concentration and so on. For colloidal systems, there will be a particular micellar concentration and also a certain aggregation concentration.

It is good that the structures of some of the ACQ dyes are shown in Figs. 5-7. This makes easier for the readers to understand the mechanism of the aggregation. The Conclusions are sustained by the presented and discussed methods and results. And, moreover, the conclusions mention also the outlook for the future planned research, as to conduct a series of in-depth studies to comprehensively showcase the unique advantages of ACQ dyes in tracking the structural integrity of anti-tumor nanocarriers.

After performing the minor revision in agreement with the above mentioned suggestions, the mini-review could be accepted for publication.

Best regards!

Author Response

Reviewer #3

The present mini-review has as main goal the subject of drug delivery systems used in different cancer therapies. This is a field of great interest nowadays, while different types of cancer are the major cause of death worldwide. Therefore, a continuous research is performed to develop better drug delivery systems, as for instance nanocarriers, described in the current manuscript.

Response: Thanks for your positive comments.

The mini-review is in general well written, the Abstract is concise and clear and the References are relevant to the field. At the key-words „antitumor” and „drug delivery” could be added. The ithenticate percentage of similarities is very low, considering the fact that it represents actually a review which show data and results already published in the literature. All the abbreviations are explained immediately when they appear for the first time and this will help thre readers. The English is fine trough the whole manuscript.

Response: Thanks for your positive comments. The keywords “antitumor” and “drug delivery” were added.

The main options currently available in general for the treatments of different cancers, are described in details: radiotherapy, anti-cancer medication and surgery. But, for a review in such a field of great interest, with a lot of recent bibliographic sources, I would expect more citations. The references list is rather short. The authors should add more references. For exampe, I did not found mentioned the osteo cancers, a type of tumor with an increasing incidence, especially for the younger population. In the literature there are several examples which proved that some organic-inorganic hybrids phosphorus containing could have a positive effect on malignant cells of human osteo-sarcoma, as for example the work of Khaled et al. https://doi.org/10.3390/cimb46050290. Such hybrid compounds containing phosphorus could be biocompatible and bioactive materials (in addition they could also contain zirconium or titanium, for example, already used in different applications in medicine), and could be synthesized by using the sol-gel method as performed in the work published by Mehring et al. https://doi.org/10.1023/A:1020797520620 , Guerrero et al. https://doi.org/10.1021/cm991125+ , Sanchez et al. https://doi.org/10.1023/A:1008753919925 , Mutin et al. https://doi.org/10.2109/jcersj2.123.709 and those are just few examples. The authors could cite them and could also find more references, especially from the last decade, relevant to the field of the present review. Moreover, the sol-gel method could be mentioned in addition to other methods reported in the review, while also the hydrogels are mentioned in the citations of the work of Zhang et al. [8 and 23].

Response: Thanks for your positive comments. The literature you recommended was quite inspiring. We agree that osteosarcoma, organic-inorganic hybrid systems and sol-gel methods should be mentioned in the text. Therefore, we have supplemented the relevant introduction where appropriate, as follows:

“In addition to the frequently reported cancer types mentioned above, there are many other cancer types that are receiving increasing clinical attention. Bone cancer, for example, is nowadays a tumor with an increasing incidence among young people.” was added.

“Researchers assessed the antitumor potential of a newly developed phosphonic acid derivative (2-carboxyethylphenylphosphonic acid) using two different in vitro human cell cultures (keratinocyte-forming cells HaCaT and osteosarcoma SAOS-2 cells), employing different techniques (MTT reagent [3-(4,5-dimethylthiazol-2-yl)-2,5-diphenyltetrazolium bromide], cytomorphometric evaluation, the lactate dehydrogenase (LDH), Hoechst staining, and the real-time reverse transcription–polymerase chain reaction (RT-PCR)), outlining its pharmacological profiles. It was found that the compounds had good biocompatibility and excellent biosafety, and their anti-osteosarcoma activity was high. Therefore, the newly developed compounds should be considered as promising candidates for further in vitro and in vivo studies related to the treatment of osteosarcoma.”

“Some phosphorus-containing organic-inorganic hybrids were demonstrated to have considerable positive effects on malignant cells of human osteosarcoma1. This phosphorus-containing organic-inorganic hybrid can be synthesized by the sol-gel method, and Merning's team found that Hybrid materials were synthesized by modification of Ti(OPr)4 by diphenyl phosphinic (DPPA) or phenyl phosphonic acid (PPA), making the material very stable2.” was added.

“It has been shown that phosphorus-containing organic-inorganic hybrids may have a positive effect on malignant cells of human osteosarcoma. The study used a variety of techniques to determine the anticancer characteristics of 2-carboxyethylphenylphosphinic acid and found that the drug has good anti-osteosarcoma activity and safety in normal cells compared to the three phosphorus-containing compounds on the market1. Typically, hybrid compounds containing phosphorus can be prepared by a sol-gel process that utilizes the ability of phosphate groups to pair with metal oxides in organic components. The composition and structural stability of the products prepared by this process were also verified3.” was added.

References:

  1. Khaled Z, Ilia G, Watz C, et al. The Biological Impact of Some Phosphonic and Phosphinic Acid Derivatives on Human Osteosarcoma. Curr Issues Mol Biol 2024;46(5):4815-31, 10.3390/cimb46050290.
  2. Mehring M, Lafond V, Mutin PH, Vioux A. New Sol-Gel Routes to Organic-Inorganic Hybrid Materials: Modification of Metal Alkoxide by Phosphonic or Phosphinic Acids. Journal of Sol-Gel Science and Technology 2003;26:99-102,
  3. Guerrero G, Mutin PH, Vioux A. Mixed Nonhydrolytic/Hydrolytic Sol−Gel Routes to Novel Metal Oxide/Phosphonate Hybrids. Chemistry of Materials 2000;12(5):1268-72, 10.1021/cm991125+.

And, as recommendations also here, please cite the references according to the guide for authors. Therefore, the names of the first 3 authors should be mentioned when a reference is cited as bibliographic source, before to mention „et al.”. So, "Zhang et al." is not correctly cited; as a consequence, you need to add the other autors and only after the third author to mention „et al.”. There are many mistakes in the references list according to the journal citations rules for authors. Please check all the list, because most of the citations mention only the first author. But, for instance, in the case of 5, 10, 11, 19, 22, 43, 44, 45, 48, 49 two authors are mentioned, and in the case of 16, 18, 36, 41, 47 three authors are mentioned. All of those are just some examples. Moreover, in some cases, the author’s initial letter of the first name is added after the family name, and in other cases before the family name (for example: Warde, U. and N.J.J.o.F. Sekar; Mukherjee, S. and P.J.C.C. Thilagar; Zhao, W. and E.M.J.A.C. Carreira; Zhao, W. and E.M.J.C.A.E.J. Carreira; Wu, L. and K.J.C.c. Burgess). The same style should be used in all cases, as recommended by the journal. Nevetheless, for the authors with more initial letters (therefore, more names) only cappital letters should be used (not N.J.J.o.F. or K.J.C.c., for instance; correctly will be N.J.J.O.F. and K.J.C.C. in those cases). Also you need to check if the DOI number is needed to be included in all the citations from the references list.

Response: Thanks for careful reviewing work. We have one-by-one double-check the format of References, and gotten rid of the issues you have pointed out.

The methods and the results are described generally in details. Still, because the main subject is related to nanocarriers as drug delivery systems and to the aggregation (as mentioned both already from the title), it would be interesting and useful for the readers to add a discussion about the aggregation in terms of colloids. The aggregatoin could occur due to DLVO and non-DLVO forces, as electrostatic forces, van der Waals, structural forces, hydration forces, micellar interactions, polymer networks if it is the case and so on. Those are theories of forces in colloidal chemistry. Are those aggregations, colloidal dispersions? If so, all of these should be discussed, in terms of charge, particles size, hydrophobic / hidrophilic character, concentration and so on. For colloidal systems, there will be a particular micellar concentration and also a certain aggregation concentration.

Response: Thanks for your valuable comments. We agree that nanoparticle aggregation due to DLVO and non-DLVO forces is a critical research topic in the field of colloidal chemistry, which deserves an in-depth investigation.

We should note that the focus of the present mini-review was Aggregation Caused Quenching (ACQ) probes with biological fate tracking properties, and thus the “aggregation” mainly refers to the aggregation of fluorescence probe (small molecular chemicals), due to conjugated aromatic structures that aggregate in aqueous biological environments due to hydrophobic interactions (e.g., π–π stacking). Therefore, DLVO and non-DLVO forces were less applicable in this scenario.

However, inspired by the authors, we admitted that the aggregation mechanisms of the probes were rather vital, and thus we performed further literature survey and added the following discussions: We added “Herein, the detailed aggregation mechanisms are as follows. AIE and ACQ probes are both organic molecules with large conjugate structures, and hence owning a high hydrophobicity. Water, in most cases, is a non-solvent for these molecules. As a result, the intermolecular interactions will take place among probes, rather than between probe and water, in an aqueous environment. In accordance with the theory of molecular exciton coupling, the plane-to-plane hypsochromically shifted hypsochromic aggregate (H-aggregate) and the end-to-end bathrochromically shifted Jelly-aggregate (J-aggregate) will generate when the probe monomers contact with each other in a non-solvent1; both H-aggregate and J-aggregate manifest in the pattern of aggregation.2” as a new paragraph after “AIE and ACQ probes represent two sides of the same coin, as both types of probes feature large, conjugated aromatic structures that aggregate in aqueous biological environments due to hydrophobic interactions (e.g., π–π stacking). The key difference lies in their molecular flexibility.”

References:

  1. Qi J, Hu X, Dong X, et al. Towards more accurate bioimaging of drug nanocarriers: turning aggregation-caused quenching into a useful tool. Adv Drug Deliv Rev 2019;143:206-25, 10.1016/j.addr.2019.05.009.
  2. Mosleh-Shirazi S, Abbasi M, Moaddeli MR, et al. Nanotechnology Advances in the Detection and Treatment of Cancer: An Overview. Nanotheranostics 2022;6(4):400-23, 10.7150/ntno.74613.

It is good that the structures of some of the ACQ dyes are shown in Figs. 5-7. This makes easier for the readers to understand the mechanism of the aggregation. The Conclusions are sustained by the presented and discussed methods and results. And, moreover, the conclusions mention also the outlook for the future planned research, as to conduct a series of in-depth studies to comprehensively showcase the unique advantages of ACQ dyes in tracking the structural integrity of anti-tumor nanocarriers.

Response: Thanks for your positive comments.

After performing the minor revision in agreement with the above mentioned suggestions, the mini-review could be accepted for publication.

Response: Thanks for your positive comments. We have substantially improved the manuscript, and hope that the current version is acceptable.
